# Characteristic Curve and Its Use in Determining the Compressive Strength of Concrete by the Rebound Hammer Test

**DOI:** 10.3390/ma12172705

**Published:** 2019-08-23

**Authors:** Dalibor Kocáb, Petr Misák, Petr Cikrle

**Affiliations:** Faculty of Civil Engineering, Brno University of Technology, Veveří 331/95, 602 00 Brno, Czech Republic

**Keywords:** rebound hammer, SilverSchmidt, concrete, compressive strength, non-destructive testing

## Abstract

During the construction of concrete structures, it is often useful to know compressive strength at an early age. This is an amount of strength required for the safe removal of formwork, also known as stripping strength. It is certainly helpful to determine this strength non-destructively, i.e., without any invasive steps that would damage the structure. Second only to the ultrasonic pulse velocity test, the rebound hammer test is the most common NDT method currently used for this purpose. However, estimating compressive strength using general regression models can often yield inaccurate results. The experiment results show that the compressive strength of any concrete can be estimated using one’s own newly created regression model. A traditionally constructed regression model can predict the strength value with 50% reliability, or when two-sided confidence bands are used, with 95% reliability. However, civil engineers usually work with the so-called characteristic value defined as a 5% quantile. Therefore, it appears suitable to adjust conventional methods in order to achieve a regression model with 95% one-sided reliability. This paper describes a simple construction of such a characteristic curve. The results show that the characteristic curve created for the concrete in question could be a useful tool even outside of practical applications.

## 1. Introduction

Concrete structures have always been built for a long service life, safety, durability, load-bearing capacity, stability, and working and functional reliability. During construction or usage, however, there may arise a need to determine or verify the properties of the concrete. This is why accurate diagnostics of concrete elements or whole structures are essential. Non-destructive testing methods (NDT) are often used for this purpose [1]. Besides the ultrasonic pulse velocity test, the rebound hammer test is a popular method, because it is easy to use and is practically non-destructive [2,3,4,5]. There are two basic ways of using it to test concrete. The first is using a rebound hammer to diagnose older structures with the primary purpose of classing the concrete according to strength or uniformity [6,7]. Testing older structures often requires removing both plaster and the top layer of concrete, as well as combining the measurement with destructive compressive strength tests performed on cores [8,9,10]. Another use of the rebound test is assessing the quality of new concrete structures, especially those with a smooth surface.

The rebound hammer test is currently the most common method of testing the hardness of concrete and was first developed in Switzerland in the 1950s as the Schmidt rebound hammer (sometimes known as the Swiss hammer) [11]. In 1950 Ernst Schmidt created the first rebound hammer, which proved to be superior to indentation methods used until that time (the most common one involved pressing a steel ball into the surface) [12]. Testing hardness with the Schmidt hammer was gradually accepted as the best, and since then concrete hardness has been determined by measuring the rebound number instead of examining the indentation produced by the ball. Today, the Schmidt hammer is the most common concrete sclerometer worldwide [13]. One of the strong points of the Schmidt hammer is its ability to test materials other than concrete (see [14,15,16,17,18,19]).

Its benefits notwithstanding, the rebound hammer test has some weak points, which should be kept in mind during testing and evaluation. Its primary purpose has always been to determine the quality of new concrete elements or structures. The general relationships between hardness and compressive strength were adjusted for this purpose, having been created by the manufacturers and then carried over to technical standards; they apply mostly to concrete of 14 to 56 days of age. The measurement range is then designed for structural concrete used during the second half of the 20th century; i.e., compressive strength of approximately 10 to 70 N/mm${}^{2}$ [20], or, more realistically, 15 to 60 N/mm${}^{2}$ [21]. When testing older concretes, it is necessary to take into account the severity of carbonation or surface damage. Kim et al. [22] concerned themselves with the influence of concrete carbonation on the rebound value and compressive strength. They discovered the depth of carbonation affects compressive strength very differently than the rebound value, which implies that the same dependencies will not apply to differently carbonated concrete. This is why they used regression analysis to create an equation that includes a factor which corrects for a reduction in compressive strength due to age, and should only be a function of the rebound value and compressive strength. The influence of concrete age and carbonation was studied in greater detail by Szilágyi et al. [11], who say that even though the rebound hammer test has been available for over 60 years, available literature lacks models that understand surface hardness as an age-dependent property. Their paper [11] presents a SBZ model; a phenomenological constructive model, which operates with concrete hardness measured by the rebound hammer test in relation to time. The SBZ model is based on the development of the capillary pore system in hardened cement paste, which, for reasons of simplicity, is replaced by the w/c ratio. The model includes the relationship between w/c ratio and 28-day compressive strength, development of compressive strength over time, relationship between compressive strength and rebound number at the age of 28 days, and the progress of carbonation depth over time and its influence on the rebound number. As with carbonation, high temperatures also affect surface hardness and compressive strength. Panedpojaman and Tonnayopas [23] found that high temperatures are detrimental to compressive strength. If concrete is exposed to fire, up to about 420 ${}^{\circ }$C the rebound number does not change in any major way, but compressive strength is lost. This is because calcium carbonate crystals form in the pore structure, causing hardness to decrease at a much slower rate and thus rendering the rebound hammer test unusable for determining compressive strength.

Correct statistical analysis of the measured values is critical for evaluating Schmidt hammer tests. Alwash et al. [24] analysed the influence of several factors on the reliability of rebound hammer test evaluation, such as within-test variability, variability of true compressive strength, number of test locations and cores used to determine the relationship between strength and the rebound number, way of choosing the test locations (random or conditional) and the model identification programme (regression or bi-objective). El Mir and Nehme [1] tested several hundred specimens focusing on the coefficient of variance in rebound values. They found that greater carbonation depth, higher w/c ratio, or higher porosity of the concrete cause the coefficient of variance to increase and reduce repeatability. However, concrete containing additions, such as metakaolin or silica fume, high-strength and high-performance self-compacting concrete, exhibit a lower coefficient of variance in rebound values. Szilágyi et al. [25] conducted an extensive statistical analysis of the variability of concrete hardness using a database of both laboratory and in situ measurements spanning over the past 60 years. The study covers several thousand tests (over eighty thousand individual rebound numbers) and shows that current sources and standards leave much to be desired in terms of the evaluation of concrete using a Schmidt impact hammer. The normality tests (the Shapiro–Wilk normality test) yielded rather inconsistent results: “the hypothesis of normality can only be accepted at very low levels of probability for individual test locations.” Given the density function of the coefficient of variance of the rebound values, a strong positive skewness can be seen. Szilágyi et al. [25] speak against directly correlating the average rebound number with compressive strength as univariate functions, and argue for a “series of multivariate functions with independent variables of the degree of hydration, type and amount of cement and aggregate, environmental conditions, and testing conditions.”

Besides assessing the quality of mature concrete or measuring its compressive strength, which, of course, carries certain associated problems, more modern types of impact hammers are capable of estimating the very early strength (also known as stripping strength) of concrete of only a few hours or days of age. This paper focuses primarily on creating a conversion formula to assess this early compressive strength, which is closely related to e.g., determining a time when it is safe to remove formwork. In fact, this is close to the original purpose of Schmidt hammers–testing concrete which is only several months old.

Many types of sclerometers have been developed over the past 70 years; however, the most common type used presently is the Schmidt type. There are several types available, differing from one another in the impact energy, shape and size of the plunger, or mechanical construction. In the past, and possibly even today, the most common Schmidt hammer is the Original Schmidt: a traditional rebound hammer, which became the basis for all the major rebound tests worldwide [26]. The basic type is the Schmidt N with impact energy of 2.207 Nm, but there is also the L type with energy of 0.735 Nm. A little over ten years ago a new model was introduced: the SilverSchmidt, which uses optical sensors to measure the impact and rebound velocity immediately before and after impact. It does not return the rebound number, but the Q-value. The different construction of the SilverSchmidt gives it the ability to measure concretes of lower as well as higher strength, and is supplied in the N and L configuration. A SilverSchmidt L can be fitted with a mushroom plunger accessory that enables measuring compressive strength as low as 5 N/mm${}^{2}$. It is designed primarily for determining concrete strength uniformity and identifying inferior areas [27]. Upon introduction to the market, the manufacturer promised better quality of measurement, but this may not be entirely true: Viles et al. [17] consider the SilverSchmidt better than its predecessor, while Szilágyi et al. [25] rank it worse than the Original Schmidt. This may be why the newest model of the Schmidt hammers can only measure the rebound number and not the Q-value: it is the Original Schmidt Live supplied with a complex application (Apple iOS and Android) for measurement, reporting, and analysis [28].

The experiment described below was performed with an Original Schmidt N, SilverSchmidt N, SilverSchmidt L, and SilverSchmidt L with a mushroom plunger (MP) accessory.

## 2. Models of Relationship between Hardness Tests and Compressive Strength

### 2.1. Common Dependence Models

This section discusses some models of dependence commonly used in contemporary civil engineering for estimating compressive strength based on rebound hammer tests. They will be further compared with models created using the experimental data.

The standard [21] defines two models of dependence between test results obtained by an Original Schmidt N and compressive strength. These are two lines for which two different ranges apply.

Line A:(1)$$f_{c}=1.75a-29,$$
where a=25−40 [-] is the rebound value and

Line B:(2)$$f_{c}=1.786a-30.44,$$
for rebound number range a=41−54 [-].

The document [29] presents dependence models for the SilverSchmidt. The relationship for test results obtained by the SilverSchmidt N is discussed in two ways: First is the median relationship with 50% reliability:(3)$$f_{c}=1.8943e^{0.064Q}$$
for Q=20−62 [-]. This curve was created based on test results obtained by the BAM institute (Federal Institute for Materials Research and Testing in Berlin, Germany) with three different kinds of concrete, which differed in the w/c ratio and type of cement, covering a strength range of $f_{c}=10-100$ N/mm${}^{2}$ [29].

Based on the results obtained by the BAM institute, The Shaanxi Province Construction Science Research Institute, China, and Hunan University, China, the document [29] defines a curve with 90% reliability
(4)$$f_{c}=2.77e^{0.048Q}$$
for a range of Q=22−75 [-]. The curve was created on the basis of recommendations in EN 13791 [30], ASTM C805 [31] and ACI 228.1 [32]. They state that the dependence model should be created so that 90% of the experimental data would lie above the curve.

The [29] also describes a model for results obtained by the SilverSchmidt L for a range of Q=20−62 [-] as
(5)$$f_{c}=1.9368e^{0.0637Q}.$$

Measurements by the SilverSchmidt L can also be made with the mushroom attachment, which should enable measurements of young concrete with low compressive strength. The relationship
(6)$$f_{c}=0.0108Q^{2}+0.2236Q$$
is defined by [27] for a range of Q=13−44 [-] and $f_{c}=5-30$ N/mm${}^{2}$.

### 2.2. Characteristic Curve

Drawing a general relationship between the rebound number and compressive strength can be a complex task. The aim of this part of the article is to show a new approach to the evaluation of NDT test results by designing the so-called characteristic curve, which ensures 95% reliability.

Several papers were published saying that formulating strength as a single parameter can be misleading [11]. Single-parameter formulation means that determining compressive strength requires knowing only the rebound number. Several publications [11,13] warn that the model of the relationship should also include e.g., the type of cement, aggregate, or w/c ratio. These and other parameters have an undeniable influence on the strength of concrete and therefore the rebound number measured by a Schmidt hammer as well.

The actual method of seeking the optimal model of relationship also requires attention. The most common method, known as regression analysis, or the least squares method, involves several assumptions, some of which may be violated during data evaluation. A determination of an optimal relationship between the rebound test and compressive strength often violates the assumption of homoscedasticity, i.e., homogeneity of variance [11]. A violation of the homogeneity of variance means that changes in the rebound number (*x* axis) change variation in compressive strength (*y* axis) as well. In other words, the higher the rebound number measured, the higher the variance in compressive strength. If the violation of homoscedasticity is ignored, the resulting model of relationship can underrepresent the value of compressive strength, especially in higher-strength concrete.

However, the experimental data presented in this paper shows that under certain circumstances the above-mentioned issues can be legitimately dismissed. These are cases where the goal is not to find a general model for every concrete, but a specific model for only one. Moreover, such a model is not intended for measuring compressive strength across the whole spectrum, but for estimating the stripping rebound number, which should correspond to stripping compressive strength (see below). Most previously published models of the relationship between the rebound number and compressive strength are designed to estimate the median value of compressive strength. The model is therefore designed as a median curve plotted through the experimental data. In theory, a model thus designed is 50% reliable: the measured value of compressive strength is 50% likely to be higher or lower.

In the civil engineering practice, however, most cases do not use 50% reliability, but 95%. Such a value of a material property is called a characteristic value. Concerning compressive strength, there is the term characteristic strength [30]. It is essentially always a 5% quantile, meaning that 95% of test results should be higher than this value. It is, therefore, a sort of one-sided interval estimate of the parameter value. Even when assessing concrete strength based on NDT results, it would often be useful to have such a one-sided estimate in the form of a curve. Points on this curve, which could be called a characteristic curve, would determine 95% one-sided interval estimates of compressive strength; i.e., characteristic strength. This section presents one of the possible ways how to construct such a curve using experimental data.

This is done using the above-mentioned method of least squares in its simplest form of a linear regression model [33]:(7)$$y=b_{1}+b_{2}\cdot x,$$
where *y* is the conventional dependent variable (compressive strength), *x* is the independent variable (rebound number) and $b_{1}$ and $b_{2}$ are the regression coefficients being determined. As will be shown later, this simple model appears suitable for all the experimental data examined here. For a better understanding of how the characteristic curve is constructed, we shall demonstrate the principle of the least squares method.

The experimental data is essentially pairs $\left(x_{i},y_{i}\right)$, where $x_{i}$ are the rebound number values and $y_{i}$ are the corresponding values of compressive strength. Further, i=1,…,n, where *n* is the number of measurements, i.e., value pairs. An important role in the least squares method is played by the matrix
(8)$$H=\left[\begin{array}{cc} n & \sum x_{i}\\ \sum x_{i} & \sum x_{i}^{2} \end{array}\right],$$
where ∑ signifies $\sum_{i=1}^{n}$, and its determinant, which can be expressed as
(9)$$\det\;H=n\sum x_{i}^{2}-{\left(\sum x_{i}\right)}^{2}.$$

The median values of regression coefficients can then be expressed using the following formulas:(10)$$b_{2}=\frac{n\sum x_{i}y_{i}-\sum x_{i}\sum y_{i}}{\det\;H},$$
(11)$$b_{1}=\bar{y}-b_{2}\bar{x},$$
where $\bar{y}$ and $\bar{x}$ are the mean values of the properties being measured [34].

At this point the model has a curve passing through the experimental data. It is generally recommended to supplement this curve with so-called confidence bands. This means a confidence band for the median curve, which is normally determined by 95 % interval estimates of $b_{1}$ and $b_{2}$, and a prediction band, which determines the 95% interval estimate of compressive strength for the given rebound number. In order to construct such bands it is necessary to express the minimum value of the sum of square errors [33,34]
(12)$$S_{min}^{*}=\sum_{i=1}^{n}{\left(y_{i}-b_{1}-b_{2}x_{i}\right)}^{2}$$
and point estimator of variance
(13)$$s^{2}=\frac{S_{min}^{*}}{n-2}.$$

Then, for any fixed *x*, it is necessary to denote the value of $h^{*}$:(14)$$h^{*}=\frac{1}{n}+\frac{n{\left(x-\bar{x}\right)}^{2}}{\det\;H}.$$

The confidence band for the median value is then determined using the formula
(15)$$\langle \left(b_{1}+b_{2}x\right)-t_{\left(1-\alpha /2\right)}s\sqrt{h^{*}};\left(b_{1}+b_{2}x\right)+t_{\left(1-\alpha /2\right)}s\sqrt{h^{*}}\rangle ,$$
where $t_{\left(1-\alpha /2\right)}$ is (1−α/2) quantile of Student’s *t*-distribution with n−2 degrees of freedom [33,34]. Next, the confidence band for the individual values (prediction band) is determined by the following formula
(16)$$\langle \left(b_{1}+b_{2}x\right)-t_{\left(1-\alpha /2\right)}s\sqrt{1+h^{*}};\left(b_{1}+b_{2}x\right)+t_{\left(1-\alpha /2\right)}s\sqrt{1+h^{*}}\rangle .$$

Both regression bands are shown in Figure 5 through Figure 11. It is useful to expand this traditional regression analysis method with regression coefficient testing and assessing the overall aptitude of the model with the multiple correlation coefficient, which in our case equals the correlation coefficient *r*. The aptitude of the model is most commonly denoted by the coefficient of determination $r^{2}$. The number $r^{2}\times 100\%$ (conventionally) signifies the percentage of $y_{i}$, which is explained by the regression model.

It is important to remember that the confidence bands are constructed as two-sided interval estimates of *y* for every fixed x. This method of denotation works for most applications of regression analysis. However, seeing as civil engineering usually works with the characteristic value (5% quantile), this method is not quite ideal.

We construct the characteristic curve using the relationship for determining the prediction band. The breadth of the band is determined by variance $s^{2}$, regression coefficients $b_{1}$ and $b_{2}$, the $h^{*}$ value for every fixed *x*, and Student’s *t*-distribution. This quantile determines, among others, the aptitude of the confidence bands, i.e., the probability with which one could expect that the true *y*-value is indeed located within this band. Adjusting the $t_{\left(1-\alpha /2\right)}$ quantile to the $t_{\left(1-\alpha \right)}$ quantile with the same degree of freedom n−2 at α=0.05 makes it possible to obtain the desired one-sided interval estimate. The characteristic curve $y_{0.05}$ can then be written as
(17)$$y_{0.05}=\left(b_{1}+b_{2}x\right)-t_{\left(1-\alpha \right)}s\sqrt{1+h^{*}}.$$

This curve essentially determines the value of compressive strength for every fixed *x*, i.e., for every fixed result of a rebound test. It is, therefore, a half-plane
(18)$$\langle +\infty ;y_{0.05}\rangle .$$

The following sections show the use of this characteristic curve for determining the characteristic stripping hardness with real-world data.

## 3. Experiment

The goal of the measurements and their evaluation was to create a new conversion relationship for determining the compressive strength of concrete by rebound hammer tests. These relationships concern mainly early-age strength: the so-called stripping strength, i.e., in the range of 5 to 10 N/mm${}^{2}$. The experiment took place in three stages.

### 3.1. Determining the Conversion Relationship for Compressive Strength of Concrete Used in Bridge Construction

The goal of the first stage was two-fold; to determine a conversion relationship for the rebound number to compressive strength for the selected concrete and to ascertain whether the Original Schmidt N and SilverSchmidt rebound hammers are suitable for this purpose: whether the statistical evaluation would confirm that the more advanced SilverSchmidt suffers from greater measurement variability [25].

The experiment required making 18 cubic specimens of 150 mm in size. The specimens were cast on-site, during the construction of an arch bridge on Svitavská street in Brno, see Figure 1. The concrete in question was C 30/37 XF4; its composition is shown in Table 1. The properties of used cement are shown in Table 2 and Table 3. Basic properties of all used aggregates (including aggregates used in Section 3.2 and Section 3.3) are shown in Table 4. The admixtures are described in Table 1 by name and further information can be found in the manufacturer’s technical data [35,36].

The specimens were made from concrete taken from three concrete mixer trucks. Six cube specimens were made from each concrete sample, see Figure 2a. The fresh-state properties, determined according to EN 12350 [37,38,39], are in Table 5, and the slump test is pictured in Figure 2b. After pouring, the specimens were covered with a PE sheet and left at the construction site for 24 h. At the age of 48 h they were transported to a laboratory at the Faculty of Civil Engineering, BUT, where they were removed from moulds and divided into 6 groups of 3 so that each group would contain a cube from truck 1, truck 2, and truck 3. The first group of 3 specimens was tested (compressive strength according to EN 12390-3 [40]) and the remaining 15 were placed under water at a temperature of (20±2)${}^{\circ }$C. The other groups were tested at the age of 3, 7, 14, 28, and 90 days. The reason the concrete was tested at different ages was to obtain data that show how the concrete’s properties develop over time. Because the concrete’s composition is known, it is possible to use the single-parameter formulation of compressive strength using the rebound number.

Each cube was measured and weighed, and afterwards mounted in a testing press and compressed with a force of 50 kN. Two opposing sides were tested for hardness with a rebound hammer. A total of 5 readings of the rebound number *R* were taken using an Original Schmidt N and 5 readings of of the *Q*-value using a SilverSchmidt N, see Figure 3. Finally, compressive strength was determined according to [40].

### 3.2. Determining the Conversion Relationship for the Stripping Strength of Precast Concrete

The goal of the second part of the experiment was to determine the relationship between the rebound number and compressive strength for concrete C 50/60, which is used in precast prestressed girders. Effort was made to capture low (stripping) strength. The formula used for making the V03 girders is detailed in Table 6. V03 girders have an asymmetrical I cross-section, length of 25.346 m, height of 1.5 m and were used in the construction of a shopping centre.

While the girders were being made, concrete C 50/60 was sampled and made into 18 cube-shaped specimens with the size of 150 mm. While still freshly poured in steel moulds, they were covered with a PE sheet and left near the girders in the manufacturing hall, where the ambient temperature did not exceed 15 ${}^{\circ }$C. Because the experiment aimed to test compressive strength at a very young age, the first three cubes were tested 27 h after cement was mixed with water. The other specimens were unmoulded at this age, placed under water and tested at the age of 42, 48, 68, 144, and 656 h. A total of 15 cubes were tested during the first 6 days and the last three at 28 days of age.

The testing was conducted similarly to the first stage of the experiment with the sole difference that besides the Original Schmidt N and SilverSchmidt N, the SilverSchmidt L both with and without the mushroom attachment was used. The SilverSchmidt L with the attachment is primarily used to test very young concrete, which is why it was used only during the first 2 days (i.e., first three measurement times), but not beyond. Figure 4 shows the manufacturing and testing of the cube specimens.

### 3.3. Determining the Conversion Relationship for the Stripping Strength of Concretes of Similar Composition

While the previous parts of the experiment were carried out with only one kind of concrete, the third involved making cube specimens from six different concrete mixtures. They only differed in the amount of cement, water, and admixtures; in essence, the w/c ratio. The actual components of the six concretes (marked I through VI) were identical. The amount of water was always balanced against the amount of admixtures so as to achieve the same workability in all the mixtures. Table 7 shows the composition of the concretes.

The basic fresh-state properties determined in compliance with [37,38,39] are detailed in Table 8.

Each concrete was used to make 6 cube specimens sized 150 mm. This made a total of 36 test specimens. While still in polyurethane moulds, the specimens were covered with a PE sheet and stored at standard laboratory conditions (ambient temperature of (20±2)
${}^{\circ }$C and humidity of (60±10)%), where they aged until testing. All the specimens were tested during the next day after being cast; the age ranged between 16 and 36 h. Immediately after unmoulding, the specimens were measured and weighed, and then mounted in a testing press and tested for the *Q*-value ten times using a SilverSchmidt PC L with the mushroom accessory attached; five measurements on two opposing sides. Finally, the compressive strength test was performed according to [40]. The goal was to determine whether it is possible to create one conversion relationship for the stripping compressive strength of several concretes of similar composition.

## 4. Results and Discussion

Test results presented in Section 3.1 (concrete bridge) and Section 3.2 (precast concrete) each time represent one concrete of known composition. The hardness and strength tests were designed so that the experimental data described the development of compressive strength over time. This justifies the single-parameter formulation of compressive strength based on the rebound number using linear regression model. The same type of model was used by the authors of the article [22], while most authors (see [3,7,10,19,44]) use rather polynomial or exponential function curves. In addition, authors of [11] refer to dozens of other publications, all of which use non-linear functions. The linear model was used in Section 3.3 as well, where there were several different kinds of concrete, but of very similar composition. These concretes varied in the amount of admixtures, aggregate, and w/c ratio, but the raw materials were the same.

The data evaluation was focused on the stripping rebound number, which should correspond to stripping compressive strength with 95% reliability; i.e., the value read from the characteristic curve (cf. Section 2.2). Stripping compressive strength, meaning strength at which formwork can be safely removed, is set here at 5 N/mm${}^{2}$.

The evaluation of results of the bridge concrete (Section 3.1) focused on the aptitude of the linear regression model for tests performed with the Original Schmidt N and SilverSchmidt N impact hammers (Figure 5 and Figure 6). The values of the coefficient of determination $r^{2}$ indicate that both these models show high aptitude, which, compared to Szilágyi’s et al. conclusions [25] is slightly higher in the case of the SilverSchmidt N. The results also show that the linear regression model used herein is more suitable for the evaluation of measurements made with a SilverSchmidt N than the exponential model proposed in [29] (see Figure 6). In both cases, a characteristic curve was created and the stripping rebound numbers evaluated. Given the range of the data obtained, approx. 30 to 80 N/mm${}^{2}$, these values are strongly extrapolated and thus are for illustration only. Table 7 summarises the results.

The results of tests described in Section 3.2 (precast concrete) are plotted in Figure 7, Figure 8, Figure 9 and Figure 10 including linear regressions. In this case the linear regression model also showed aptitude, exhibiting high values of the coefficient of determination $r^{2}$ (see Table 9). The results of tests performed with an Original Schmidt N on this particular concrete show that relationships described in [21] may undervalue compressive strength (see Figure 7). The non-linear character of the dependence of rebound tests performed with a SilverSchmidt N, L, and L-MP [27,29] was not confirmed.

The greatest aptitude of the linear model was demonstrated by test results for the similar concretes described in Section 3.3 ($r^{2}=0.963$). These tests were performed with a SilverSchmidt L with the mushroom plunger accessory. As with the previous cases, these results indicated no need to use an exponential or other model.

## 5. Conclusions

The experiments indicate the following:The paper describes an innovative design of constructing a characteristic curve, which can be used to find the stripping rebound number obtained by different types of rebound hammers. Using the characteristic curve appears suitable for finding a specific rebound number which would indicate, with 95% reliability, that the concrete has the required stripping strength (in this paper it is 5 N/mm${}^{2}$).The data shows that the concretes tested herein (including concretes with different formulas but the same raw materials) do not require a regression model more complex than a simple line. Moreover, this simplest shape of regression model enables a relatively simple formulation of the characteristic curve.We certainly do not question the validity of relationships described elsewhere (Section 2.1), or their ability to represent the relationship between the rebound number and compressive strength. It should also be remembered that these general models were created for concretes of virtually any composition, and a simple line would not be enough.Our regression models and rebound numbers representing stripping strength always apply to the one specific concrete they were designed for and do not work with others. The goal of the paper was to show how such regression models may be created for any concrete.

## Figures and Tables

**Figure 1 materials-12-02705-f001:**
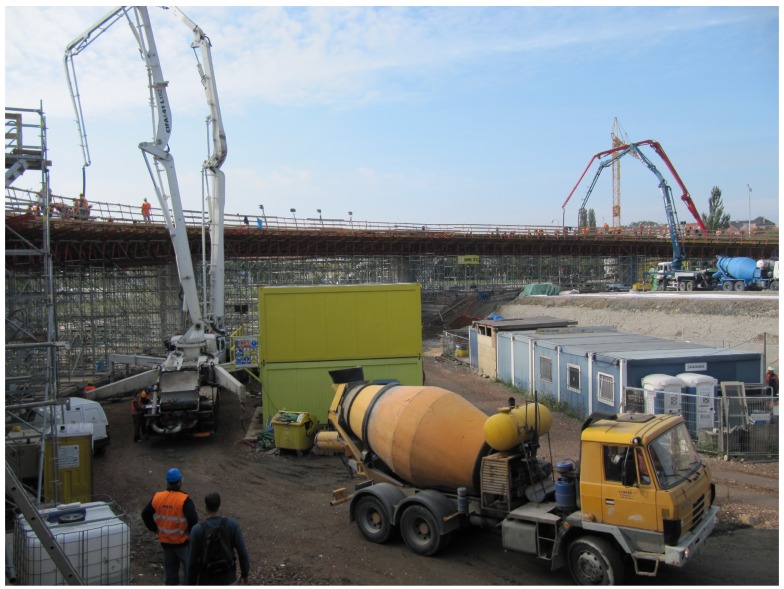
Bridge construction during which concrete C 30/37 XF4 was sampled.

**Figure 2 materials-12-02705-f002:**
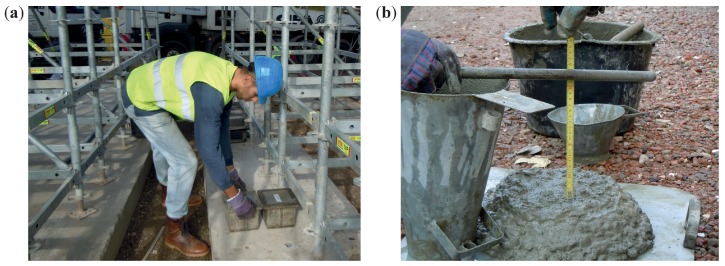
(**a**) cube specimens being made; (**b**) consistency test-slump.

**Figure 3 materials-12-02705-f003:**
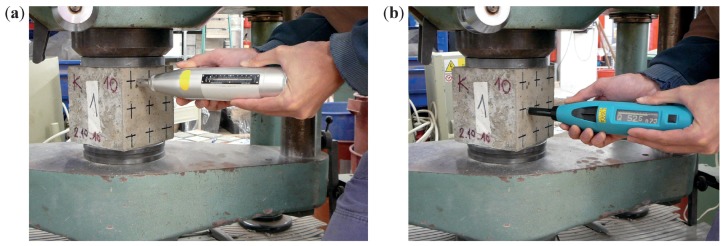
(**a**) testing a cube with an Original Schmidt N; (**b**) with a SilverSchmidt N.

**Figure 4 materials-12-02705-f004:**
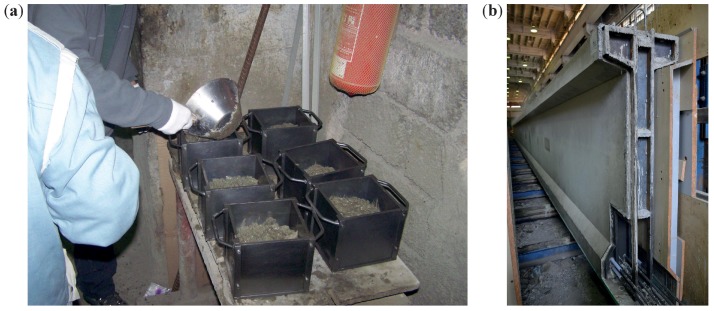
(**a**) making of the specimens; (**b**) manufacture of the V03 girders.

**Figure 5 materials-12-02705-f005:**
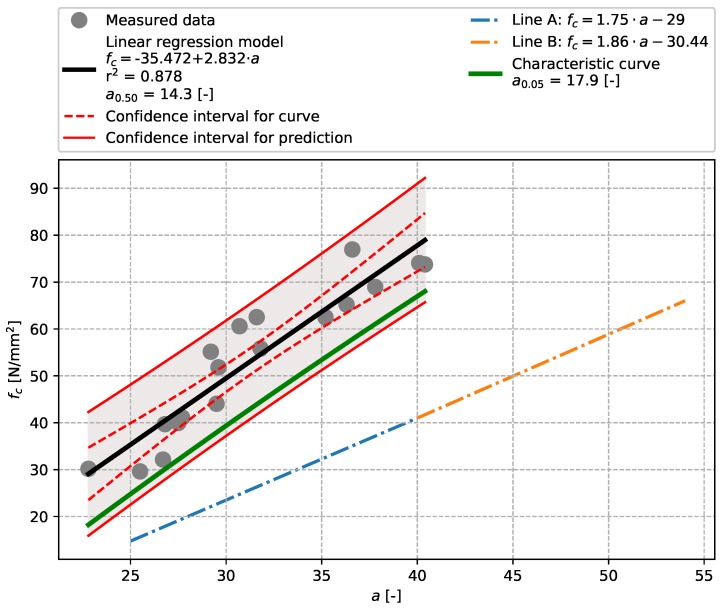
Bridge concrete described in Section 3.1—model of the dependence of the rebound number [-] vs. compressive strength $f_{c}$ [N/mm${}^{2}$]—Original Schmidt N.

**Figure 6 materials-12-02705-f006:**
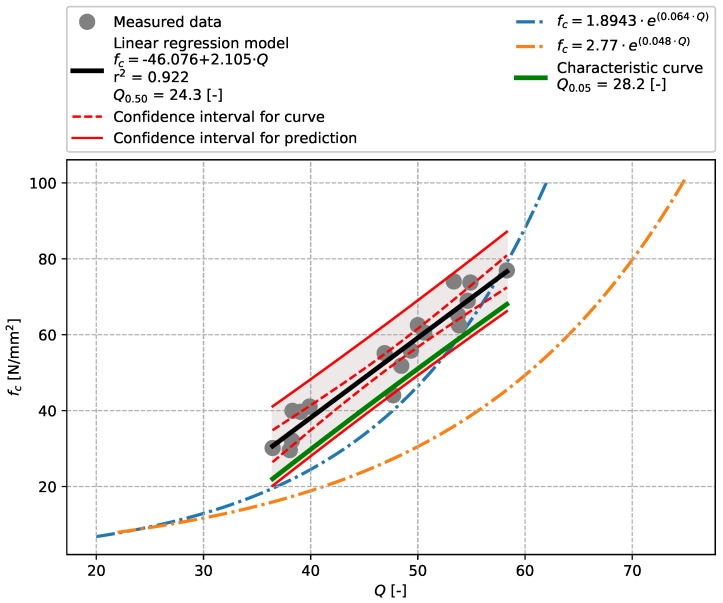
Bridge concrete described in Section 3.1—model of the dependence of the *Q*-value [-] vs. compressive strength $f_{c}$ [N/mm${}^{2}$]—SilverSchmidt N.

**Figure 7 materials-12-02705-f007:**
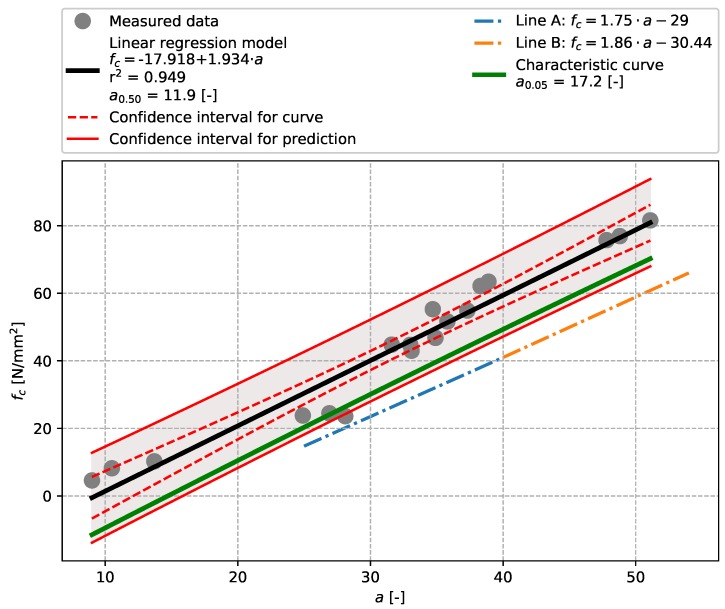
Precast concrete described in Section 3.2—model of the dependence of *a* [-] vs. compressive strength $f_{c}$ [N/mm${}^{2}$]—Original Schmidt N.

**Figure 8 materials-12-02705-f008:**
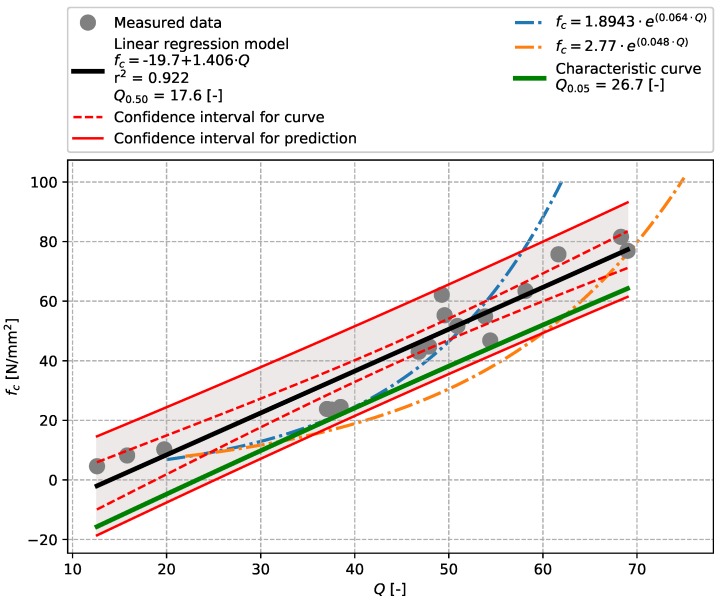
Precast concrete described in Section 3.2—model of the dependence of *Q*-value [-] vs. compressive strength $f_{c}$ [N/mm${}^{2}$]—SilverSchmidt N.

**Figure 9 materials-12-02705-f009:**
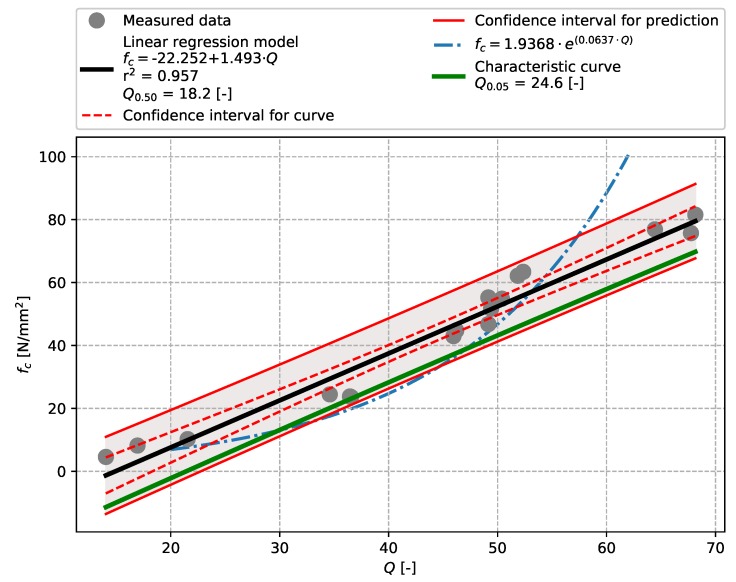
Precast concrete described in Section 3.2—model of the dependence of *Q*-value [-] vs. compressive strength $f_{c}$ [N/mm${}^{2}$]—SilverSchmidt L.

**Figure 10 materials-12-02705-f010:**
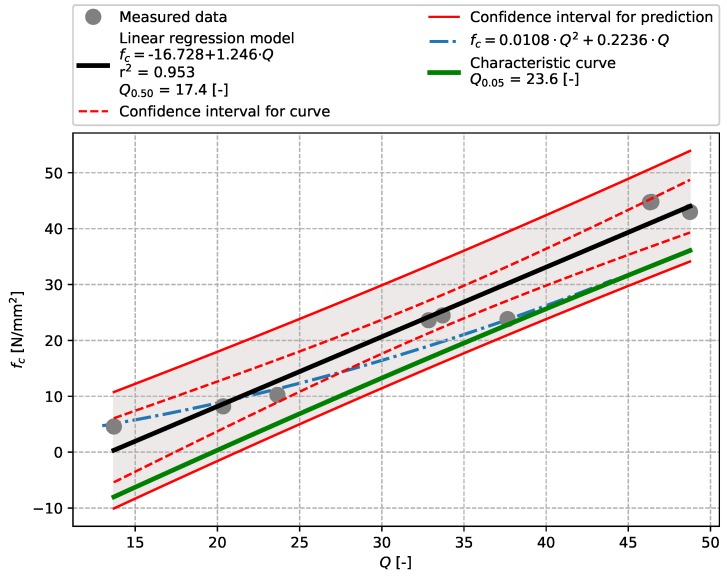
Precast concrete described in Section 3.2—model of the dependence of *Q*-value [-] vs. compressive strength $f_{c}$ [N/mm${}^{2}$]—SilverSchmidt L—MP.

**Figure 11 materials-12-02705-f011:**
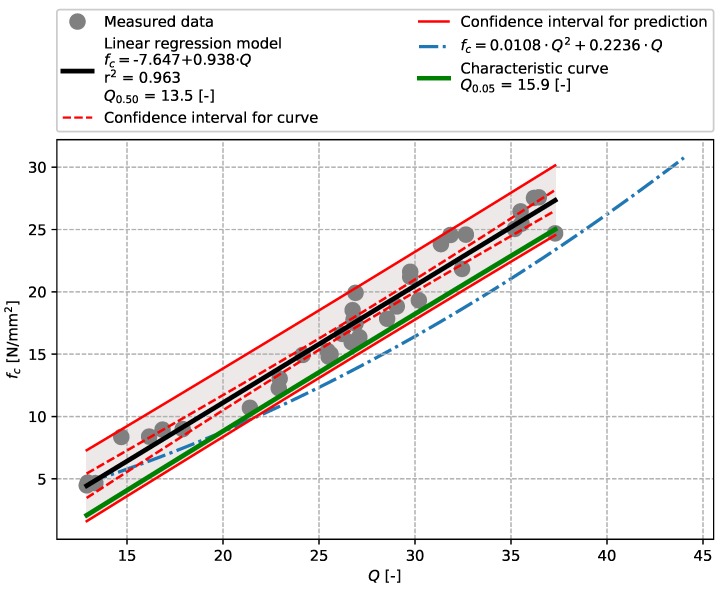
Precast concrete described in Section 3.3—model of the dependence of *Q*-value [-] vs. compressive strength $f_{c}$ [N/mm${}^{2}$]—SilverSchmidt L – MP.

**Table 1 materials-12-02705-t001:** Composition of concrete C 30/37 XF4.

Component	Amount (kg Per 1 m${}^{3}$)
Cement CEM I 42.5 R (Mokrá cement plant)	400
Aggregate 0-4 mm—Ledce (see Table 4)	700
Aggregate 8-16 mm—Olbramovice (see Table 4)	669
Aggregate 11-22 mm—Lomnička (see Table 4)	284
Water	172
Air-entraining admixture—Sika${}^{®}$ Aer 200	0.60
Plasticiser—Sika${}^{®}$ ViscoCrete${}^{®}$-5-800 Multimix (AT)	2.40

**Table 2 materials-12-02705-t002:** Chemical properties of Cement CEM I 42.5 R (Mokrá cement plant) according to EN 196-2 [41].

Component/Property	Average Value (%)
CaO	63.7
SiO${}_{2}$	19.6
Al${}_{2}$O${}_{3}$	4.8
Fe${}_{2}$O${}_{3}$	3.3
MgO	1.4
SO${}_{3}$	3.1
Cl${}^{-}$	0.040
K${}_{2}$O	0.75
Na${}_{2}$O	0.19
Na${}_{2}$O Equivalent	0.69
Insoluble residue	0.7
Loss of ignition	3.4

**Table 3 materials-12-02705-t003:** Basic physical and mechanical properties of Cement CEM I 42.5 R (Mokrá cement plant) according to EN 196-6 [42] and EN 196-8 [43].

Parameter	Average Value
Blain (m${}^{2}$/kg)	408
Density (kg/m${}^{3}$)	3110
Heat of hydration (7 days) (J/g)	310

**Table 4 materials-12-02705-t004:** Basic properties of used aggregates.

Aggregate	Type	Rock	Specific Density (Mg/m${}^{3}$)	Loose Bulk Density (Mg/m${}^{3}$)
Ledce	Natural quarried	Gravel sand	2.553	1.408
Lípa	Natural quarried	Gravel sand	2.583	1.508
Bratčice	Natural quarried	Gravel sand	2.610	1.537
Olbramovice	Natural crushed	Granodiorite	2.640	1.450
Lomnička	Natural crushed	Gneiss	2.690	1.530
Litice	Natural crushed	Granodiorite	2.700	1.600

**Table 5 materials-12-02705-t005:** Fresh-state properties of concrete C 30/37 XF4.

Mix Truck	Slump Test (mm)	Air Content (%)	Bulk Density (kg/m${}^{3}$)
1	240	4.2	2320
2	180	3.8	2370
3	220	3.9	2360

**Table 6 materials-12-02705-t006:** Composition of concrete C 50/60.

Component	Amount (kg Per 1 m${}^{3}$)
Cement CEM I 42.5 R (Mokrá cement plant)	450
Aggregate 0-4 mm—Lípa (see Table 4)	690
Aggregate 4-8 mm—Litice (see Table 4)	215
Aggregate 8-16 mm—Litice (see Table 4)	845
Water	180
Plasticiser—Stachement 2180	4.50

**Table 7 materials-12-02705-t007:** Composition of the concretes.

Component	Amount (% Per 1 m${}^{3}$)					
	I	II	III	IV	V	VI
Cement CEM I 42.5 R (Mokrá cement plant)	87.50	100.00	87.50	100.00	87.50	100.00
Aggregate 0/4 mm—Bratčice (see Table 4)	218.75	206.25	218.75	206.25	218.75	206.25
Aggregate 4/8 mm—Olbramovice (see Table 4)	46.25	46.25	46.25	46.25	46.25	46.25
Aggregate 8/16 mm—Olbramovice (see Table 4)	173.75	173.75	173.75	173.75	173.75	173.75
Water	44.75	44.75	43.50	43.50	41.00	41.00
Plasticiser—Sika${}^{®}$ ViscoCrete${}^{®}$ 4035	0.220	0.250	0.438	0.500	0.438	0.500
Air-entraining admixture—Sika${}^{®}$ LPS A 94	0.000	0.000	0.000	0.000	0.188	0.188

**Table 8 materials-12-02705-t008:** Composition of the concretes.

Property	Concrete					
	I	II	III	IV	V	VI
Bulk Density (kg/m${}^{3}$)	2300	2300	2270	2300	2190	2260
Slump test (mm)	50	60	70	50	60	50
Air content (%)	2.8	3.2	3.5	3.0	6.2	5.7

**Table 9 materials-12-02705-t009:** Summary.

Rebound Hammer	Concrete Type	$Q_{0.50}\;\left(a_{0.50}\right)$	$Q_{0.05}\;\left(a_{0.05}\right)$	$r^{2}$	Figure
Original Schmidt N	Bridge concrete	14.3	17.9	0.878	Figure 5
SilverSchmidt N	Bridge concrete	24.3	28.2	0.922	Figure 6
Original Schmidt N	Precast concrete	11.9	17.2	0.949	Figure 7
SilverSchmidt N	Precast concrete	17.6	26.7	0.922	Figure 8
SilverSchmidt L	Precast concrete	18.2	24.6	0.957	Figure 9
SilverSchmidt L–MP	Precast concrete	17.4	23.6	0.953	Figure 10
SilverSchmidt L–MP	Concretes of similar composition	13.5	15.9	0.963	Figure 11

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
