# Peer review of "Characteristic Curve and Its Use in Determining the Compressive Strength of Concrete by the Rebound Hammer Test"

_materials, 2019, doi:10.3390/ma12172705_

Round 1
Reviewer 1 Report
Authors have performed a detailed study for the numerical evaluation of rebound hammer test results.
This could produce a very good technical report but, to my opinion, it is not suitable for a scientific journal. Its rather a good technical report than a paper
Due to this reason, I finally do not proceed to a detailed review (such as much up of references).
Reviewer 2 Report
The authors of submission Materials-549020 present an interesting work on the developing a simple construction of a characteristic curve to estimate the compressive strength from the results of rebound hammer test. The topic is of interest in that it has the potential to be used in engineering practice to improve the accuracy of the experimental methods in charactering material properties.
The paper is very well-structured, readable, and coherent. Hence, I would recommend for the acceptation in the present form.
Reviewer 3 Report
The article is interesting for the engineering environment, has the potential to use the results in building practice.
The intoduction and theoretical part takes up a lot of space (230 lines) compared to the experimental part (116). It is necessary to shorten the Introduction and develop the Discussion.
Table 1 - if the particle size of aggregates has been entered correctly? - Aggregate 8-16 mm (natural crushed) and aggregate 11-22 mm (natural crushed).
In addition, the characteristics used to make concrete materials were not given in the experimental part. Characteristics of cement, coarse aggregate, admixtures were not given. It must be completed. Please specify the cement composition, properties, specific and bulk density of aggregates, similarly with air-entraining admixture and plasticizer. In the journal Materials, it is strictly observed.
The discussion of results should be developed and related to literature. The authors referred only to the studies of Szilágyi, K. et al. - item [26]. The remaining references are the standard and The SilverSchmidt Reference Curve. Please, expand the discussion.
Literature - please check the edition. It seems to me that there are double spaces before the names of journals.
Round 2
Reviewer 1 Report
Authors' work is a very good technical report (case study), especially after their last additions.
Such reports are after every case that includes use on non-destructive tests for concrete. Authors also admit that "the goal of the paper was to show how such regression models may be created for any concrete". This is evident.
In order that this report could expand to a scientific paper, the proposed model should been also verified by sets of data that have not been already used for its compilation. In such a case, major revision would be needed.
Author Response
Response to Reviewer 1 Comments
Point 1: Authors' work is a very good technical report (case study), especially after their last additions.
Such reports are after every case that includes use on non-destructive tests for concrete. Authors also admit that "the goal of the paper was to show how such regression models may be created for any concrete". This is evident.
In order that this report could expand to a scientific paper, the proposed model should been also verified by sets of data that have not been already used for its compilation. In such a case, major revision would be needed.
Response 1: We respect the opinion of the opponent, but we still think that the text can be published as a scientific article, which the other two reviewers are convinced. The paper brings innovative solutions and a new approach to regression modelling and therefore significantly exceeds the standard case study or technical report. We have re-arranged the order of sections 2.1 and 2.2 to make it more obvious that creating a characteristic curve is our new solution.
Reviewer 3 Report
Thank you for the corrections.
Author Response
Thank you very much.